# Moderate nucleotide diversity in the Atlantic herring is associated with a low mutation rate

**Chungang Feng[1][†], Mats Pettersson[1][†], Sangeet Lamichhaney[1][†], Carl-Johan Rubin[1], Nima Rafati[1], Michele Casini[2], Arild Folkvord[3,4], Leif Andersson[1,5,6]***

[1]Science for Life Laboratory, Department of Medical Biochemistry and Microbiology, Uppsala University, Uppsala, Sweden; [2]Department of Aquatic Resources, Institute of Marine Research, Swedish University of Agricultural Sciences, Lysekil, Sweden; [3]Department of Biology, University of Bergen and the Hjort Center of Marine Ecosystem Dynamics, Bergen, Norway; [4]Institute of Marine Research, Bergen, Norway; [5]Department of Animal Breeding and Genetics, Swedish University of Agricultural Sciences, Uppsala, Sweden; [6]Department of Veterinary Integrative Biosciences, Texas A&M University, College Station, United States

**Abstract** The Atlantic herring is one of the most abundant vertebrates on earth but its nucleotide diversity is moderate (π = 0.3%), only three-fold higher than in human. Here, we present a pedigree-based estimation of the mutation rate in this species. Based on whole-genome sequencing of four parents and 12 offspring, the estimated mutation rate is $2.0 \times 10^{-9}$ per base per generation. We observed a high degree of parental mosaicism indicating that a large fraction of these de novo mutations occurred during early germ cell development. The estimated mutation rate – the lowest among vertebrates analyzed to date – partially explains the discrepancy between the rather low nucleotide diversity in herring and its huge census population size. But a species like the herring will never reach its expected nucleotide diversity because of fluctuations in population size over the millions of years it takes to build up high nucleotide diversity.

***For correspondence:** leif.
andersson@imbim.uu.se

[†]These authors contributed
equally to this work

**Competing interests:** The
authors declare that no
competing interests exist.

**Reviewing editor:** Molly
Przeworski, Columbia University,
United States

## Introduction

Empirical observations of nucleotide diversity in different species show that the variation is often much smaller than would be expected from simple population genetic models (*Leffler et al., 2012*). The Atlantic herring (*Clupea harengus*) is a good example of the paradox, since, in spite of an enormous census population size about $10^{12}$ (*Supplementary file 1*), its nucleotide diversity (π = 0.3%) (*Martinez Barrio et al., 2016*) is middle-of-the-road when compared to terrestrial mammals, e.g. 0.1% for humans (*Hernandez et al., 2011*) and 0.9% for European rabbits (*Carneiro et al., 2014*) with much smaller census populations. A large census population does not necessarily mean that the long term effective population size ($N_e$) is large but the extremely low genetic differentiation at selectively neutral loci between geographically distant populations strongly suggests that current $N_e$ must be high and genetic drift very low in the Atlantic herring (*Martinez Barrio et al., 2016*).

Before the NextGenerationSequencing-era, mutation rates were estimated by comparative genomics, by relating sequence differences to fossil record-dated estimates of species divergence times, or by tracking changes at specific loci in experimental studies. However, since species divergence is hard to date and the use of a small subset of loci can introduce bias, these methods have limited accuracy (*Drake et al., 1998*). More recently, affordable whole genome sequencing has facilitated two approaches to estimate mutation rates: mutation accumulation lines and parent-offspring

**eLife digest** Evolution by natural selection favours the survival of individuals that are well suited to their environment. This process depends on genetic differences between individuals that make some more able to survive than others. These genetic differences are the result of mutations in DNA of germ-line cells, that is, the cells that produce egg cells and sperm. These mutations mean that new offspring always have a few small differences in some of the genes they inherited from each of their parents.

DNA contains strings of molecules known as bases. These act as individual "letters" in the genetic code of an individual. Rapid sequencing of DNA to find out the order of these bases makes it possible to study the rate of mutations within a species. This provides a way to measure how different an individual is from its parents and, by extension, the potential of the species to diversify and adapt to different environments. There are over a trillion Atlantic herring in the Atlantic Ocean, so this fish is an ideal model to study the effects of germ-line mutations on genetic diversity. In 2016, a group of researchers reported that there is relatively little genetic diversity across Atlantic herring. Given the large population, this suggested that the mutation rate in this species may be low.

Feng, Pettersson, Lamichhaney et al. – who were also involved with the earlier work – sequenced the DNA of two families of Atlantic herring raised in captivity to calculate the rate of germ-line mutations in this species. The results showed that, on average, two changes occur per one billion letters in the genetic code in each generation. That is one to two new mutations per egg cell or sperm. This is the lowest mutation rate yet recorded in any animal with a backbone and is around six times lower than the mutation rate in humans.

Whilst the low mutation rate in Atlantic herring means there are few differences between individual fish, the extremely large number of these fish on the planet still means that there is enough diversity across the population to allow the species to adapt to changing conditions. This work is important for conservation as it highlights the great variation in potential genetic diversity across species. Future work will need to examine why the mutation rate in Atlantic herring is so low and compare it more widely to mutation rates in other species.

comparisons. The mutation accumulation approach, where an inbred line is maintained for a number of generations and the mutation rate is measured by counting up differences between the first and last generation, has the advantage of scalability, since it is possible to increase the number of mutation events observed by including more generations. On the other hand, the approach requires an organism that can be reproduced as viable inbred lines, and it is difficult to fully eliminate purifying selection against deleterious new mutations. The parent-offspring approach, which relies on using high coverage whole-genome sequencing to detect differences between parents and their offspring, alleviates the cultivation related issues, and has thus become the preferred method for estimating the mutation rate in non-model organisms. The trade-off is that the total number of mutation events per progeny will typically be small.

Currently, the number of studies using any of the methods outlined above remains small, and the available data is somewhat biased towards unicellular organisms (*Dettman et al., 2016*; *Dillon et al., 2015*; *Farlow et al., 2015*; *Lee et al., 2012*; *Ness et al., 2012*; *Zhu et al., 2014*), insects (*Keightley et al., 2014*, *2015*, *2009*) and mammals (*Harland et al., 2016*; *Kong et al., 2012*; *Uchimura et al., 2015*; *Venn et al., 2014*), while including a single plant (*Ossowski et al., 2010*) and one bird (*Smeds et al., 2016*). In all, this leaves large sections of the tree of life essentially unexplored. This is problematic for drawing general conclusions about the relationship between neutral diversity, effective population size and mutation rate, which is a topic of considerable interest in population genetics (*Leffler et al., 2012*; *Lynch, 2010*).

In this study, to our knowledge the first of its kind in a teleost, we estimate the genome-wide point mutation rate in Atlantic herring. The Atlantic herring was chosen due to its suitability as a population genetic model system; it is one of the most abundant vertebrate species on earth with external reproduction involving large numbers of gametes per reproducing adult. In essence, these

properties make the Atlantic herring one of the best approximations of a randomly mating, infinite size population among vertebrates. In addition, there exists a high-quality draft genome assembly (*Martinez Barrio et al., 2016*), which is a pre-requisite for a study of this kind. We have employed the parent-offspring approach, and base our measurement on two families, each containing two parents and six offspring. We here estimate the spontaneous mutation rate to be $2.0 \times 10^{-9}$ per site per generation in the Atlantic herring, six-fold lower than the rate in humans and the lowest rate reported so far for a vertebrate.

## Results

### Whole genome sequencing and variant calling

We have generated two-generation experimental pedigrees for spring-spawning Atlantic and Baltic herring (classified as a subspecies of the Atlantic herring by *Linnaeus (1761)*), each comprising the two parents and six offspring (*Table 1*). We performed whole-genome sequencing of these two pedigrees using genomic DNA isolated from muscle tissue. As detection of de novo mutations requires high sequence coverage, we sequenced each individual to ~45–71 x (*Table 1*), in line with the procedures used in previous studies (*Keightley et al., 2015*; *Kong et al., 2012*). The sequences were aligned to the recently published Atlantic herring genome (*Martinez Barrio et al., 2016*). A total of 5.3 (Atlantic) and 5.2 (Baltic) million raw SNPs were detected in each pedigree, respectively, using GATK (see Materials and methods) (*McKenna et al., 2010*).

### Identification and validation of the de novo mutations

Detection of de novo mutations with high confidence requires a careful examination of raw variant calls and application of highly stringent filtering criteria. Using a standard genotype-calling pipeline will typically lead to the great majority of novel sequence variants detected being false positives. Screening of provisional candidate mutations in a single offspring indicated that this was the case, as many candidates could not be verified using Sanger sequencing. Hence, in order to minimize the

**Table 1.** Summary of the pedigrees used for whole-genome sequencing.

| No | ID | Pedigree | Sequencing depth (x) | De novo mutations |
|----|----|----------|----------------------|-------------------|
| Pedigree 1, Atlantic herring | | | | |
| 1 | AM8 | Father | 65.7 | N.A. |
| 2 | AF8 | Mother | 70.2 | N.A. |
| 3 | AA1 | Offspring | 65.6 | 1 |
| 4 | AA2 | Offspring | 70.9 | 2 |
| 5 | AA3 | Offspring | 47.2 | 0 |
| 6 | AA4 | Offspring | 66.9 | 3 |
| 7 | AA5 | Offspring | 64.2 | 4 |
| 8 | AA6 | Offspring | 61.2 | 1 |
| Pedigree 2, Baltic herring | | | | |
| 9 | BM19 | Father | 71.8 | N.A. |
| 10 | BF21 | Mother | 65.1 | N.A. |
| 11 | BB1 | Offspring | 74.5 | 2 |
| 12 | BB2 | Offspring | 61.6 | 1 |
| 13 | BB3 | Offspring | 75.0 | 0 |
| 14 | BB4 | Offspring | 69.9 | 2 |
| 15 | BB5 | Offspring | 60.6 | 2 |
| 16 | BB6 | Offspring | 62.6 | 1 |

N.A. = Not available.

frequency of false positives by the de novo calls using only the GATK variant caller, we separately performed variant calling using SAMTOOLS (*Li et al., 2009*) and only selected novel mutations detected by both variant callers (*Figure 1*). In addition, we applied strict filtering criteria in order to remove variants detected due to sequencing and alignment errors. We excluded variant calls from genomic regions with low mappability (see Materials and methods) and repetitive regions detected by Repeat Masker (*Smit et al., 2013*). Furthermore, we defined the cut-off parameters for sequence depth, SNP and genotype quality-related statistics using the set of SNPs that were fixed for different alleles in both parents and thus heterozygous in all offspring (*Figure 1*, Materials and methods). As this strict filtering could lead to failure to detect some fraction of true heterozygotes, we estimated the false negative rate of our pipeline by calling SNPs in each individual offspring separately, in order to eliminate bias stemming from shared SNPs present in multiple individuals being called with higher power. For this analysis we used 116,910 polymorphic sites where the parents were homozygous for different alleles in the joint genotype calling. The expectation is that these sites are heterozygous in all offspring, but that information did not influence SNP calling. By separating the individuals, we mimicked the situation for de novo mutations, which are typically not shared. Using the same pipeline as for the de novo detection, the average detection rate of such heterozygous positions across all offspring was 94.1%, yielding a false negative rate of 5.9%. As an alternative way of estimating the false negative rate, we used a simulation procedure where we generated mutated reads for 1000 positions within callable regions. Each site in each offspring had its frequency of mutated reads determined by a sample from the observed frequency distribution of called heterozygous sites in the original data set (see Materials and methods). Across all offspring, we found an overall frequency of 2.7% false negative calls, while roughly 9% of sites failed to generate a call (*Supplementary file 2*). Overall, the two methods used are in agreement. However, for the purpose of the final calculations we will use the empirical estimate of 5.9%, which includes both incorrect and failed calls, as it is derived directly from the real data set. The choice has minor effects on the estimated mutation rate, as using the simulated value would result in the final rate being approximately 5% higher.

This stringent filtering procedure identified a total of 17 candidate de novo mutations, nine in the Atlantic pedigree and eight in the Baltic one (*Tables 1* and *2*). Two of the 17 de novo mutations were each found in two different offspring from the same pedigree.

We performed Sanger sequencing of the genomic regions around each of these putative de novo mutations in all parents and the 12 offspring (*Figure 1—figure supplement 1*). This confirmed that all 17 putative de novo mutation events were genuine and all the peak ratios of two alleles were close to 1:1 consistent with germ-line mutations. Thus, we did not observe any false positives in this study.

In order to estimate the transmission frequencies of our detected de novo mutations, we measured the rate of transfer of the de novo mutations in a larger set of offspring (*n* = 46 and 50 per family), in order to infer when during the formation of the parental germ line the mutation occurred (*Table 2*). For eight out of seventeen de novo mutations we observed more than one sibling carrying exactly the same mutation (*Table 2*). The range of occurrences for the de novo mutations was one to nine among the 50 offspring. Even the maximum of the observed transfer rates (18% for scaffold 153: 2,684,380 T>G) was significantly lower than the 50% expected for a fixed mutation (p=1.4×$10^{-3}$, Fisher's exact test). About half of the de novo mutations were present in two or more offspring, indicating that they occurred during early germ cell divisions. Assuming that the number of cell divisions from zygote to mature sperm or egg is similar in Atlantic herring to the one in mammalian species, we can conclude from a recent simulation study (*Harland et al., 2016*) that it would be highly unlikely to observe such a high rate of parental mosaicism unless a large fraction of the de novo mutations occurred during early germ cell divisions. Further, the incidence of parental mosaicism differed significantly between the two families included in this study (*Table 2*; p=0.01, Fisher's exact test). The finding that the same mutation was observed in two or more siblings for eight of the putative de novo mutations confirms that these must be germ-line mutations and not somatic mutations.

## Parental origin of de novo mutations

We also explored if the 17 germ line de novo mutations had a paternal or maternal origin. For 14 of the de novo mutations, we could detect an additional segregating site within the same Illumina

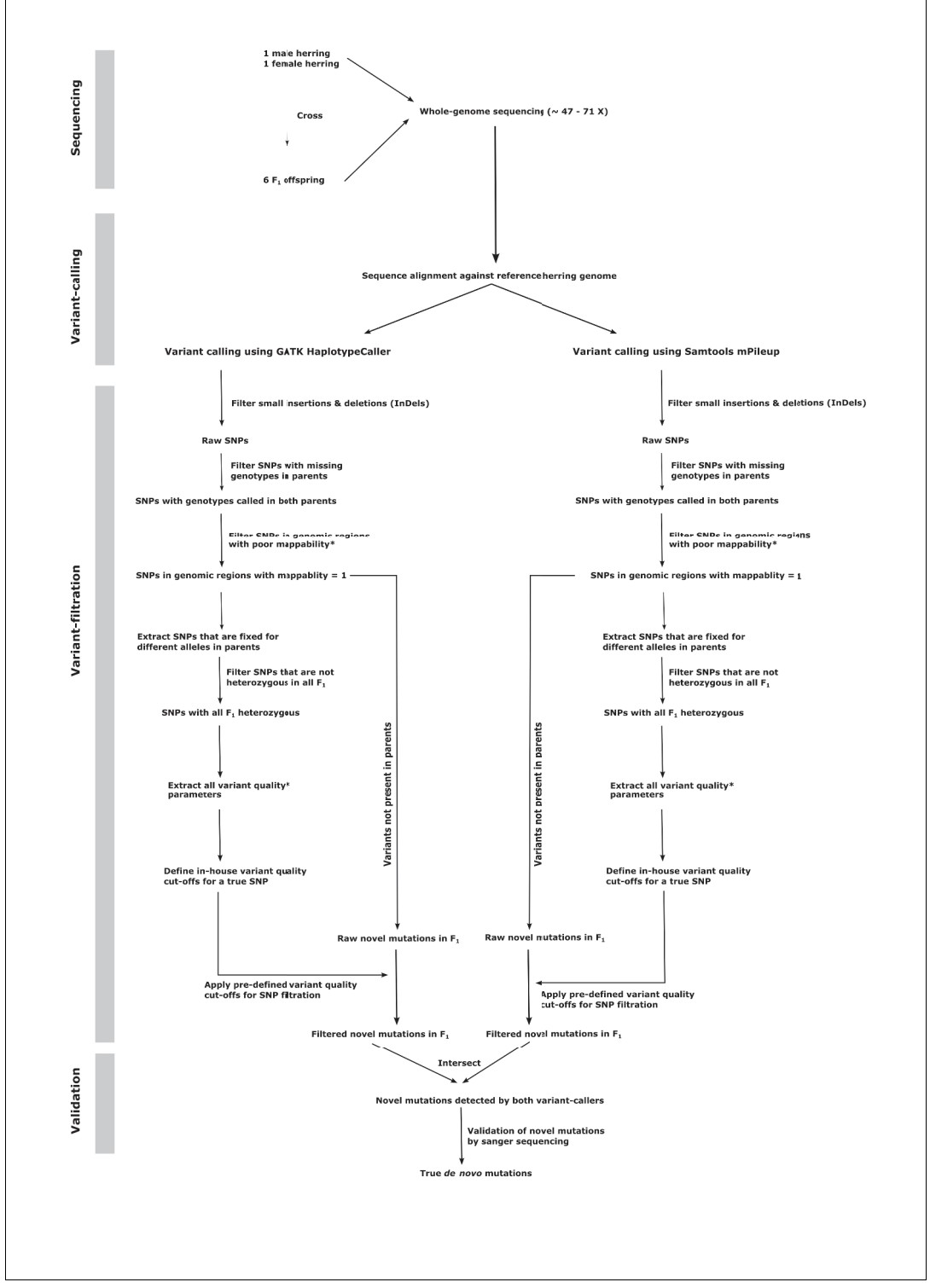

**Figure 1.** Flowchart describing the de novo mutation-calling pipeline. A schematic illustration of the steps used in calling and filtering the candidate mutations.

The following figure supplement is available for figure 1:

**Figure supplement 1.** Sanger sequencing chromatograms of the de novo mutations.

**Table 2.** Summary of the de novo mutations identified in Atlantic herring.

| SNP position | | Mutation | | | | | |
| --- | --- | --- | --- | --- | --- | --- | --- |
| Scaffold:position | Id | Ref | Var | Freq[†] | Origin[‡] | Type[§] | Region |
| 1157:174,127 | AA4 | T | A | 1/50 (-) | M | TV | Intergenic |
| 153:2,684,380 | AA2 | T | G | 9/50 (18%) | P | TV | Intronic |
| 241:7,752,158 | AA5 | C | A | 5/50 (10%) | M | TV | Intergenic |
| 4:5,098,858 | AA5 | T | C | 2/50 (4%) | M | TS | Intronic |
| 481:1,927,799 | AA4, AA5* | C | A | 6/50 (12%) | P | TV | 3' UTR |
| 61:815,077 | AA4 | A | T | 3/50 (6%) | N.A. | TV | Intergenic |
| 62:613,919 | AA1, AA6* | C | A | 6/50 (12%) | M | TV | Intergenic |
| 729:1,499,224 | AA2 | C | T | 4/50 (8%) | M | TS | Intronic |
| 887:195,946 | AA5 | G | A | 1/50 (-) | P | TS | Intronic |
| 10:1,443,002 | BB4 | C | T | 1/46 (-) | P | TS | Intronic |
| 151:267,875 | BB5 | A | T | 1/46 (-) | P | TV | Exonic |
| 177:1,045,894 | BB1 | A | G | 1/46 (-) | P | TS | Intronic |
| 194:478,776 | BB6 | A | G | 1/46 (-) | N.A. | TS | Intronic |
| 246:1,890,479 | BB4 | T | C | 1/46 (-) | P | TS | Intergenic |
| 257:380,993 | BB2 | G | A | 1/46 (-) | M | TS | Intergenic |
| 26:2,976,192 | BB1 | T | C | 2/46 (4%) | P | TS | Intronic |
| 37:1,374,669 | BB5 | G | A | 1/46 (-) | M | TS | Intronic |

*Same mutation detected in two progeny.

[†]Number of siblings carrying the de novo mutation; - the frequency of transmission was only estimated when two or more progeny with the de novo mutation was detected.

[‡]M:Maternal, P:Paternal, N.A. = Not available

[§]TV = Transversion, TS = Transition

sequencing read (length 125 bp) or mate-pair read that spanned the respective de novo mutation and was uniquely associated with either parent. In these cases, the parental origin could be directly inferred. We were able to infer the parental origins of one additional de novo mutation by PCR cloning and sequencing.

Out of the 15 mutations for which their parental origin was determined, there was no significant difference between paternal (eight) and maternal (seven) mutations (*Table 2*). A paternal bias in the origin of de novo mutations has been shown in mammals, such as human (ratio = 3.9) (*Kong et al., 2012*) and chimpanzee (ratio = 5.5) (*Venn et al., 2014*), where the main reason is thought to be the larger number of cell divisions during spermatogenesis than during oogenesis (*Crow, 2000*). While the numbers are small, a binomial test against the human ratio indicates that the gender bias in herring, if it exists at all, is significantly weaker than in humans (p=0.004). In herring, both sexes produce large numbers of gametes and males only produce sperm during the spawning season (a few months per year). Furthermore, the high degree of parental mosaicism indicates that a large fraction of the de novo mutations reported here must have occurred during early germ cell development when we do not expect a strong gender effect. These circumstances offer a reasonable explanation to the balanced parental origin of de novo mutations in the Atlantic herring.

## Characteristics of de novo mutations

Among the 17 de novo mutations, there were 10 transitions and seven transversions, yielding a transition/transversion ratio of 1.4, which can be compared with a genome-wide ratio of 1.1 for previously reported SNPs (*Martinez Barrio et al., 2016*). An overrepresentation of transitions is expected, and the observed ratio falls in the range found in previous de novo mutation studies. For example, Kong *et al.* identified 3344 transitions out of 4933 events (ratio = 2.1) in humans (*Kong et al., 2012*), while Keightley *et al.* found five out of nine events (ratio = 1.25) in the tropical

butterfly *Heliconius melpomene* (*Keightley et al., 2015*). In humans and other mammals there is a well-established excess of CpG>TpG mutations (*Kong et al., 2012*). CpG methylations also occur in teleosts (*Rai et al., 2010*), but in our dataset only 1 out of 17 de novo mutations was of this type. This frequency (6%) is below, but not significantly different from the frequency reported for human (19%) (*Kong et al., 2012*) (binomial test, p=0.06).

There were six mutations located in intergenic regions, nine intronic mutations, one 3' UTR mutation and one exonic mutation. In all, this is a distribution that does not deviate significantly from random expectation, given the composition of the genome after mappability filtering (p=0.65, Fisher's exact test).

## Estimation of mutation rates

We identified nine and eight de novo mutations in the Atlantic herring and the Baltic herring pedigrees, respectively. Since we had 12 progeny in total and two of the mutations were detected twice among the sequenced progeny, our estimate of the de novo mutation rate is 0.79 (19/24). After strict filtering of genomic regions with low mappability and repetitive sequences, we had ~442 Mb of sequence available for variant screening. Based on the distribution of read coverage in a random subset of the genome, we estimated that 2.6% of this region have insufficient depth for successful SNP calling, giving us a final callable region of 442 × 0.974 = 431 Mb (representing ~51% of the genome). The mutation rate per site per generation can thus be estimated as 19/ ($2 \times 12 \times 431 \times 10^6$)=$1.8 \times 10^{-9}$ (95% CI = 1.1–2.7 × $10^{-9}$, assuming that the mutations are Poisson distributed). If we correct for the estimated false negative rate (5.9%) we obtain: 2.0 × $10^{-9}$ (95% CI = 1.1–2.9 × $10^{-9}$).

Based on historical sampling of several herring stocks, we estimated the minimum generation time of Atlantic herring before the onset of large-scale commercial fishing to be approximately six years (*Supplementary file 3*). Using this historical generation time, the mutation rate per site per year in the Atlantic herring was estimated at 3.3 × $10^{-10}$ (95% CI = 1.9×$10^{-10}$ – 4.8 × $10^{-10}$).

## Discussion

This study provides new insights regarding factors affecting the mutation rate and levels of nucleotide diversity in vertebrates. Our finding of a high degree of parental mosaicism for the detected de novo mutations is consistent with several recent studies indicating that the early cleavage cell divisions in the germ-line are particularly mutation-prone (*Harland et al., 2016*; *Rahbari et al., 2016*; *Ségurel et al., 2014*). A high rate of de novo mutations at early germ-cell divisions has also been reported for *Drosophila* (*Gao et al., 2014*).

The estimated mutation rate (μ = 2.0×$10^{-9}$) for the Atlantic herring is the lowest for a vertebrate species to date (*Table 3*); about six-fold lower than in humans. It should be noted that this number reflects the rate in the callable fraction of the genome, which by definition does not contain repeat regions. Thus, the true genomic average could be somewhat higher, as replication of repetitive regions tends to be more error-prone, but the decreased calling power in those regions makes diversity hard to estimate in an unbiased fashion. However, these issues are not unique to the Atlantic herring, similar caveats apply to estimates of mutation rates in other species as well, and the results should thus be comparable across species. In this study we surveyed about 51% of the current genome assembly for the Atlantic herring and we used our previously published population data (*Martinez Barrio et al., 2016*) to estimate the nucleotide diversity in the parts of the genome that were included and excluded in the current study to address the concern that we may have underestimated the mutation rate because the rate is higher in the part that was excluded. This analysis showed that the nucleotide diversities in the excluded and included parts were almost identical (π = 0.00318 and π = 0.00304, respectively). In conclusion, this analysis does not indicate a major difference in mutation rates between the two parts of the genome.

By combining the now estimated mutation rate with the neutral diversity level (π = 0.0032) found by *Martinez Barrio et al. (2016)* and the expected relationship between nucleotide diversity, the mutation rate and effective population size ($N_e$) for selectively neutral alleles (π = 4 $N_e$ μ), we obtain an estimated $N_e$ of 4 × $10^5$. While this number is larger than for most terrestrial animal species, it is still much lower than census population size of the herring, about $10^{12}$ (*Supplementary file 1*). There are several factors that may contribute to this discrepancy, but demographic history stands

**Table 3.** Summary of mutation rates measured to date.

| Species | Taxonomic group | $\mu$ | Method* | Genome size (Mb) | $N_e$† |
|---|---|---|---|---|---|
| *Pseudomonas aeruginosa* | Bacteria | $7.9 \times 10^{-11}$ | MA[1] | 6.3 | $2.1 \times 10^8$ |
| *Burkholderia cenocepacia* | Bacteria | $1.3 \times 10^{-10}$ | MA[2] | 8.1 | $2.5 \times 10^8$ |
| *Escherichia coli* | Bacteria | $2.2 \times 10^{-10}$ | MA[3] | 4.6 | $1.6 \times 10^8$ |
| *Chlamydomonas reinhardtii* | Unicellular eukaryotes | $2.1 \times 10^{-10}$ | MA[4] | 120 | $7.8 \times 10^7$ |
| *Saccharomyces cerevisiae* | Unicellular eukaryotes | $1.7 \times 10^{-10}$ | MA[5] | 12.2 | $1.2 \times 10^7$ |
| *Schizosaccharomyces pombe* | Unicellular eukaryotes | $2.1 \times 10^{-10}$ | MA[6] | 12.6 | $1.4 \times 10^7$ |
| *Arabidopsis thaliana* | Plants | $7.1 \times 10^{-9}$ | MA[7] | 119 | $2.8 \times 10^5$ |
| *Pristionchus pacificus* | Invertebrates | $2.0 \times 10^{-9}$ | MA[8] | 133 | $1.8 \times 10^6$ |
| *Caenorhabditis elegans* | Invertebrates | $1.5 \times 10^{-9}$ | MA[9] | 100 | $5.2 \times 10^5$ |
| *Caenorhabditis briggsae* | Invertebrates | $1.3 \times 10^{-9}$ | MA[9] | 108 | $2.7 \times 10^5$ |
| *Drosophila melanogaster* | Invertebrates | $3.2 \times 10^{-9}$ | MA[10] PO[11] | 144 | $1.4 \times 10^6$ |
| *Heliconius melpomene* | Invertebrates | $2.9 \times 10^{-9}$ | PO[12] | 274 | $2.1 \times 10^6$ |
| *Daphnia pulex* | Invertebrates | $5.7 \times 10^{-9}$ | MA[13] | 250 | $8.2 \times 10^5$ |
| Atlantic herring (*Clupea harengus*) | Teleosts | $2.0 \times 10^{-9}$ | PO* | 850 | $4.0 \times 10^5$ |
| Collared flycatcher (*Ficedula albicollis*) | Birds | $4.6 \times 10^{-9}$ | PO[14] | 1118 | $2.0 \times 10^5$ |
| Mouse (*Mus musculus*) | Mammals | $5.4 \times 10^{-9}$ | MA[15,16] | 2808 | $1.8 \times 10^5$ |
| Cattle (*Bos taurus*) | Mammals | $9.7 \times 10^{-9}$ | PO[17] | 2725 | $3.7 \times 10^4$ |
| Chimpanzee (*Pan troglodytes*) | Mammals | $1.2 \times 10^{-8}$ | PO[18] | 3231 | $2.9 \times 10^4$ |
| Human (*Homo sapiens*) | Mammals | $1.2 \times 10^{-8}$ | PO[19] | 3236 | $2.4 \times 10^4$ |

*MA = Mutation Accumulation, PO = Parent-Offspring. The values are from the following sources: 1. **Dettman et al. (2016)**; 2. **Dillon et al. (2015)**; 3. **Lee et al. (2012)**; 4. **Ness et al. (2012)**; 5. **Zhu et al. (2014)**; 6. **Farlow et al. (2015)**; 7. **Ossowski et al. (2010)**; 8. **Weller et al. (2014)**; 9. **Denver et al. (2012)**; 10. **Keightley et al. (2009)**; 11. **Keightley et al. (2014)**; 12. **Keightley et al. (2015)**; 13. **Keith et al. (2016)**; 14. **Smeds et al. (2016)**; 15. **Lindsay et al. (2016)**; 16. **Uchimura et al. (2015)**; 17. **Harland et al. (2016)**; 18. **Venn et al. (2014)**; 19. **Kong et al. (2012)**.

†$N_e$ is calculated as $\pi/4\mu$. The underlying $\pi$ estimates are all from **Lynch et al. (2016)** except for herring (present study), collared flycatcher (**Ellegren et al., 2012**) and cattle (**Daetwyler et al., 2014**).

out as the most prominent factor. Using coalescent analysis and allele frequency distributions, *Martinez Barrio et al. (2016)* showed that the herring population is expanding from a previous bottleneck. Since the diversity-based estimate of effective population size can be considered as an average over time this bottleneck still have a major impact on the current nucleotide diversity. Population genetics theory implies that it will take $4N_e$ generations before populations reach their genetic equilibrium (*Kimura and Ohta, 1973*). We have estimated the generation interval to approximately six years in this study (*Supplementary file 3*) and a conservative estimate of the current (not long-term) $N_e$ is $10^7$, which appears reasonable since we estimated long-term $N_e$ at $4 \times 10^5$ and we have evidence for population expansion (e.g. excess of rare alleles (*Martinez Barrio et al., 2016*)). These figures indicate that it will take about 240 million years before the herring populations reach genetic equilibrium. Thus, it is obvious that a species with a large population size like the herring and a relatively long generation interval will never reach genetic equilibrium. Background selection (the elimination of deleterious alleles) and selective sweeps will also lead to reductions in nucleotide diversity at linked neutral sites (*Gillespie, 2000, 2001*). Furthermore, highly efficient purifying selection decreases the fraction of the genome that appears as selectively neutral (*Ohta, 1973*) which is also expected to lead to a slightly reduced nucleotide diversity.

The fact that the observed mutation rate is unusually low in the Atlantic herring is of interest in relation to the drift-barrier hypothesis (*Lynch et al., 2016*), which predicts that the purging of slightly deleterious mutations affecting the mutation rate is particularly effective in species that have a very large effective population size, large fecundity and close to random mating, conditions which the Atlantic herring meets (*Table 3*). However, since the population size of the Atlantic herring

appears to have fluctuated over time (*Martinez Barrio et al., 2016*), it remains unclear exactly how powerful selection has been in a time-averaged perspective, which means the support for the drift-barrier hypothesis is not unconditional. Additionally, the low body temperature of a marine fish may also slow down the metabolic rate which has been suggested to decrease the mutation rate (*Martin and Palumbi, 1993*). In a recently released study, *Malinsky et al. (2017)* used three trios representing three species of Lake Malawi cichlids and estimated the overall mutation rate to 3.5 $\times$ $10^{-9}$ compared with 2.0 $\times$ $10^{-9}$ for the Atlantic herring. However, these fish both have a lower estimated effective population size than herring (*Malinsky et al., 2017*) and live in warmer waters. In conclusion, there is still a need to compare our data with mutation rates from additional species, with lower populations sizes but similar body temperatures, before we can draw firm conclusions about the relationship between population size and mutation rate.

According to simple, ideal-case population genetic models there should be a positive relationship between nucleotide diversity and population size, so that a population at mutation-drift balance has a nucleotide diversity of $4N\mu$. However, as outlined above, this expectation is disrupted by population size fluctuations over time and selective forces. In practice, population sizes are only weakly, if at all, correlated with nucleotide diversity (*Leffler et al., 2012*). Our finding that the inherent mutation rate is approximately six times lower in Atlantic herring than in humans indicates that differences in intrinsic mutation rate is also an important factor when comparing nucleotide diversities among species. In the case of the Atlantic herring, the low mutation rate, the demographic history and efficient positive and negative selection, all contribute to explaining the apparent disparity between nucleotide diversity and the census population size in the Atlantic herring.

## Materials and methods

### Sample

Two full-sib families were generated by crossing wild-caught Atlantic herring from Bergen (Norway) and Baltic herring from Hästskär (Sweden). For each family, six offspring from a total of 50 progeny were selected for sequencing together with the two parents. Our aim was to determine the mutation rate to its order of magnitude and one to two significant digits. Thus, a samples size of 12 progeny was expected to result in about 100 detectable novel mutations based on previously known vertebrate mutation rates and the size of the genomic regions we could use to detect mutations. Genomic DNA was isolated from muscle tissue using Qiagen DNeasy Blood and Tissue kit. DNA libraries were constructed using the TruSeq PCR-free kit.

### Whole-genome sequencing

All individuals were sequenced on Illumina HiSeq2500 machines, using 2 $\times$ 125 bp paired reads to a sequencing depth of ~47–71X (*Table 1*). The short reads were aligned to the *Clupea harengus* reference genome (*Martinez Barrio et al., 2016*) using BWA v0.6.2 (*Li and Durbin, 2009*) with default parameters. The data were then filtered based on mappability, calculated using GEM (*Derrien et al., 2012*), within the reference assembly, so that only positions with mappability 1 that were also inside 1 kb windows with average mappability >0.95 were included in the downstream analysis; 442 Mb (52%) of genome sequence passed this filtering step. The sequence data have been deposited in the SRA archive (PRJNA356817).

### Variant calling and filtration

Sequence alignments from the previous step were used for calling variants using two separate tools; GATK v3.3.0 (*McKenna et al., 2010*) and SAMTOOLS v.1.19 (*Li et al., 2009*). We used GATK HaplotypeCaller with default parameters that performs simultaneous calling of SNP and Indels via local de novo assembly of haplotypes (see GATK manual for details). We ran HaplotypeCaller separately for each individual to generate intermediate genomic VCF (*Danecek et al., 2011*) files (gVCF). Afterwards, we used the GenotypeGVCFs module in GATK to merge gVCF records from each individual (altogether 12 from the two pedigrees) using the multi-sample joint aggregation step that combines all records, generate correct genotype likelihood, re-genotype the newly merged record and re-annotate each of the called variants and thereby generate a VCF file. For SAMTOOLS, we used the

standard multi-sample SNP calling pipeline (*Li et al., 2009*) using the 'mpileup' module for calling raw variants.

Once we got the raw variant calls, we filtered small insertions and deletions and only used SNPs for downstream analysis. Furthermore, we also removed SNPs that had missing genotypes in one or both parents, as these SNPs were not informative. Afterwards, we extracted a subset of SNPs where parents were homozygous for different allele and all six offspring were heterozygous (the genotype calls were considered heterozygous in offspring if the minor allele frequency was >25%). The SNP quality annotations in this set of 'known' heterozygous offspring were used as proxy to consider the quality parameter of true SNPs in the dataset. We extracted various SNP quality annotations recorded in the VCF file like total read depth, mapping quality, mapping quality rank sum, base quality, base quality rank sum, read position rank sum, quality by depth, genotype quality, allele depth (see GATK manual for details on these parameters) and examined their distributions in the subset of our known heterozygous offspring. As these quality parameters were close to being normally distributed, we used the threshold of mean ±2 x standard deviation for each of these quality estimates as the standard cut-offs for our in-house SNP filtering pipeline to filter raw SNPs in our entire dataset (*Figure 1*).

## De novo mutation calling

From the filtered SNP dataset generated in the previous step, we further selected those sites where both parents were homozygous for the reference allele and at least one offspring carried the variant allele in the heterozygous state. These two sets of raw novel mutations in offspring independently called by GATK and SAMTOOLS were then intersected and the sites that were detected by both variant callers were considered as our true de novo mutations among the progeny.

## Experimental validation and parental origin

PCR amplification and Sanger sequencing of both strands verified all candidate mutations. We inferred the parental origin of the de novo mutations based on flanking SNP alleles that could be verified by Sanger sequencing and only have been transferred from one of the parents. The parental origin of fourteen de novo mutations could be directly deduced from SNP alleles segregating between the two parents present on the same short Illumina read and mate-pair read as the de novo mutation (at least 5 reads). The parental origin of one additional de novo mutations was determined via cloning PCR fragments and sequencing; we sequenced at least 7 independent clones for each de novo mutation.

## Estimation of the false negative rate

Firstly, we estimated the false negative rate by performing genotype calls at those nucleotide positions where the parents were fixed for different alleles. The genotype calls for progeny were done without using the information for parents to mimic the detection of de novo mutations. Secondly, we also used simulation to estimate the false negative rate. From the previously determined callable fraction of the genome, we selected approximately 1000 sites without any existing polymorphism for each offspring and then introduced de novo mutations. Then, we aligned the new reads and called SNPs using the pipeline described in *Figure 1*. Finally, we compared the SNP calls with expected genotypes based on the mutated sites and calculated the false negative rate.

## Estimation of generation time

The generation length of populations with overlapping generations is equal to the mean age of parents (*Hill, 1979*). Following *Miller and Kapuscinski (1997)*, this was approximated as the mean age of spawners (age-specific number of fish multiplied by the age-specific proportion of reproductive fish) weighted by age-specific mean weights. In our analyses we used age-specific weights as proxy for age-specific fecundity, since in Atlantic herring weights and fecundity are strongly and nearly linearly correlated (*Arula et al., 2012*; *Oskarsson and Taggart, 2006*). We estimated the generation time for the herring stocks with data starting shortly after the end of the World War II, a period characterized by still low commercial exploitation which started to increase after the early 1960s. The stocks were the North Sea/Skagerrak/Kattegat/English Channel, the Celtic Sea, the West of Scotland/West of Ireland, the Irish Sea and the Norwegian spring spawning herring. Data on age-

specific abundance, maturity and mean weight were extracted from stock assessment reports (*ICES, 2015*, *ICES, 2016*).

The generation time was very similar for almost all the stocks in the first available period after the World War II, characterized by low exploitation, i.e. in 1947-1965. During this period, the generation time declined between ~6 years in late 1940s (corresponding to the lowest exploitation) and ~5 years in 1965, decreasing further in successive years. No data were available for the period before 1947 when the generation time was likely to have been higher. The Norwegian spring spawning herring showed a higher generation time than the other stocks, oscillating around 10 years in the 1950s. We therefore consider the generation time of 6 years as a minimum estimate for Atlantic herring under no or moderate exploitation.

## Acknowledgements

We thank Michel Georges for comments on the manuscript and Maurice Clarke for assistance in collating the data for the estimation of herring generation time. The work was funded by the ERC project BATESON (to LA), the GENSINC project funded by the Norwegian Research Council (to AF and LA). Illumina sequencing was performed by the SNP&SEQ Technology Platform, supported by Uppsala University and Hospital, SciLifeLab and Swedish Research Council (80576801 and 70374401). Computer resources were provided by UPPMAX, Uppsala University.

## Additional information

### Funding

| Funder | Grant reference number | Author |
|---|---|---|
| European Research Council | Bateson | Leif Andersson |
| Norges Forskningsråd | 254774 | Arild Folkvord Leif Andersson |

The funders had no role in study design, data collection and interpretation, or the decision to submit the work for publication.

### Author contributions

CF, Formal analysis, Investigation, Writing—original draft, Writing—review and editing; MP, SL, Formal analysis, Validation, Investigation, Methodology, Writing—original draft, Writing—review and editing; C-JR, Validation, Investigation, Methodology; NR, Validation, Investigation, Methodology, Writing—review and editing; MC, Investigation, Methodology; AF, Resources, Funding acquisition; LA, Conceptualization, Supervision, Funding acquisition, Investigation, Methodology, Writing—original draft, Project administration, Writing—review and editing

### Author ORCIDs

Chungang Feng, http://orcid.org/0000-0002-7031-4211
Sangeet Lamichhaney, http://orcid.org/0000-0003-4826-0349
Leif Andersson, http://orcid.org/0000-0002-4085-6968

## Additional files

### Supplementary files

• Supplementary file 1. Estimated population size of major stocks of Atlantic herring in the North East Atlantic Ocean including the Baltic Sea.

• Supplementary file 2. Summary statistics of the SNP calls underlying the estimation of the false negative rate by means of simulation.

• Supplementary file 3. Estimates of generation time for different stocks of herring.

## Major datasets

The following dataset was generated:

| Author(s) | Year | Dataset title | Dataset URL | Database, license, and accessibility information |
|---|---|---|---|---|
| Feng C, Pettersson M, Lamichhaney S, Rubin CJ, Rafati N, Casini M, Folkvord A, Andersson L | 2017 | Moderate nucleotide diversity in the Atlantic herring is associated with a low mutation rate | http://www.ncbi.nlm.nih.gov/bioproject/356817 | Publicly available at NCBI BioProject (accession no: PRJNA356817) |

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
