## [Decision Letter]

Thank you for submitting your article "Moderate nucleotide diversity in the Atlantic herring is associated with a low mutation rate" for consideration by *eLife*. Your article has been favorably evaluated by Diethard Tautz (Senior Editor) and three reviewers, one of whom is a member of our Board of Reviewing Editors. The following individual involved in review of your submission has agreed to reveal his identity: Charles Baer (Reviewer #3).

The reviewers have discussed the reviews with one another and the Reviewing Editor has drafted this decision to help you prepare a revised submission.

Given the enthusiasm for the central finding – of a lower mutation rate in a species with a large census population size – we would like to give you the opportunity to revise your manuscript and resubmit it for review. Please note, however, that reviewers are requesting both an additional analysis to confirm your estimated false negative rate and a thorough revision of the interpretation and discussion of the results. Specifically, in revising your manuscript, we ask that you address these following points:

1) Reviewers 1 and 2 raised important concerns about the lack of details provided in the Materials and methods. Moreover, they would like to see the false negative rate confirmed by a second approach (a number of possible options are suggested in their specific comments below).

2) Reviewers 1 and 2 felt that claims about the effective population size of herring are often incorrectly or confusingly framed. Notably, there is clear circularity in some sections, since N_e_ estimates from diversity rely on mutation rate estimates. This section needs to be thoroughly rewritten and, as reviewer 1 suggests, could be reframed in terms of the census population size instead. Indeed, Lewontin's paradox is about how similar diversity levels are across species with vastly different census population sizes.

This issue affects not only the interpretation of the findings in herring, but also the Discussion of findings of other organisms (Table 3). This Table reports N_e_ estimates and direct mutation rate estimates, but it is unclear if the N_e_ estimates were obtained using the direct estimates of mutation rates. The authors should really be starting with diversity levels, measured in a fairly uniform way across species, then using the direct mutation rate estimates to obtain long-term estimates of N_e_ – and only then examining the dependence of mutation rates with Ne.

3) Reviewers 1 and 2 also both doubted how much support the data really provide for Lynch's hypothesis, given the data for different species presented by the authors in Table 3 (even taken at face value).

*Reviewer #1:*

I find the central result – the estimated mutation rate per generation-to be of great interest, even though I have a concern about the estimated false negative that I would like to see addressed (see 1 below). I also found the results about mosaicism striking and think they warrant further emphasis in the paper. However, I found the discussion of effective population sizes very confusing, and suggest that this section be removed or thoroughly rewritten. I am also unconvinced by how clearly the data support Lynch's hypothesis, when considered together with other data (presented in Table 3).

1) Given that the mutation rate reported by the authors is lower than ever report for another multi-cellular organism, a reliable estimate of the false negative is critical. I found the approach taken by the authors sensible, but would like to see a second approach to be convinced by the finding that it is indeed very low. Options include relaxing the threshold and verifying these additional mutation calls, to ensure that few validate, or flipping alleles in the reads to simulate new mutations, as done for example by Venn et al. 2014 Science.

In this regard, I note that the authors were only able to assess mutations in roughly half of the genome. That is not a problem in itself, but it does mean that they are making a strong assumption that the mutation rate in the other half of the genome is the same. In this regard, it would be helpful to know whether diversity levels are lower in the fraction of the genome surveyed. Regardless of the answer, a caveat is needed about the rest of the genome and what it may yet reveal.

On a more minor note, I would like to know how many sites were used to estimate the false negative rate by their approach.

2) Quite confusingly, the authors refer repeatedly to an estimate of N_e_ obtained "from coalescent analyses" without mentioning that this estimate depends entirely on assumptions about the mutation rate. From what I gathered looking at the previous paper, the authors used the human per generation rate for that analysis (?!), suggesting that they should actually be reporting 7x that number for N_e_. Moreover, as best as I can understand, they are referring to the estimate of N_e_ at present, not the long-term estimate. Yet neutral diversity levels (i.e., the 0.3% per site that they cite) represent much longer time periods (4N_e_ generations on average in a constant population size), and they estimate huge fluctuations in N_e_ over that time scale in their previous paper.

In fact, the only information that they have about N_e_ that does not depend on the mutation rate is that LD decays quickly (similar to the scale of *Drosophila*, not humans), which could be due to a high rate of recombination, and that Fst is low, which could be due to high migration. Thus, I would argue that they don't really know if N_e_ "should be high", as argued throughout.

What they do know, and in my mind do not discuss enough, is that the census population size is surely huge, on the order of billions per school. So the mystery is how N_e_ can be only 0.3x10^-2^/(4x1.7x10^-9^) =4.4x10^[5]^ when the census population size is probably on the order of 10^[10]^or more. That is in my view quite a striking result, but needs to be recast as such.

3) Similarly, in Table 3, the authors present a set of N_e_ estimates without stating whether these are obtained assuming the direct estimates of mutation rates or not. With regard to Lynch's hypothesis, these data are not overwhelmingly supportive, other than in suggesting that mammals, who are fewer in number at least, have a slightly higher mutation rate. In part, I would argue that reflects the limitations of N_e_ as a proxy for the efficiency of selection, since it has different meanings in inbreeding species than outcrossers, and structured populations than panmictic ones. More striking in my mind is how *similar* rates are across very diverse organisms.

As an aside, the existing mice estimates are missing (http://biorxiv.org/content/early/2016/10/20/082297 and Ishimura et al. 2015 Genome Research.

Additional comments:

The authors might add a p-value to the statement that the male bias is less than in mammals (e.g., assuming α = 3, I obtain p=0.01).

In discussing the fraction of CpG transitions, it may be helpful to add a comment about whether CpGs are methylated in teleosts.

Are there estimates of the mutation rate based on the number of substitutions between species? How do they compare to the estimate presented here?

Table 2 suggests apparent difference in the degree of mosaicism between the two families (e.g., I obtain p = 0.04 by a FET). The authors should comment on this point.

*Reviewer #2:*

Feng et al. provide the first direct estimation of mutation rate in fish and bring up the possibility that the low mutation rate they observe is directly related to the low nucleotide diversity in herring, despite its incredibly large population size. I find this study of great interest in expanding our understanding of mutation rates and their variation but had some issues in how the authors interpreted their data, methodological approaches and the level of detail provided in the manuscript. These are outlined below:

1) The manuscript contains too little detail on approach to evaluate the methods. Several filters are referenced in the text such as sequence depth, SNP and genotype quality statistics, but no information is given on what these parameters were. The authors state that the cutoffs were based on SNPs that were fixed for different alleles in both parents but it is unclear how – did the authors use the mean values from this distribution, the minimum? Furthermore, the authors state that they follow the GATK pipeline to generate variant calls but there are different paths through this pipeline depending on the specifics of the data, for example, whether known variants are available. Information about library preparation (e.g. confirming that libraries were PCR-free) would also be useful. In my view, the manuscript cannot be appropriately evaluated without much of this missing information. As a minor note, I looked up the SRA accession to try to get more detail on the library preparation and could not find it on the SRA (perhaps the link is still private?).

2) Evaluation of the false negative and number of background sites. The authors use a sensible approach to evaluate their false negative rate among SNPs by focusing on sites that are homozygous for alternative alleles in both parents and asking how frequently these are detected as heterozygous in the offspring. However, I have several questions/concerns about the false negative rate and number of background sites.

A) False negative rate: Because the false negative rate is a fundamentally important in the major result of this study, I would like to see a secondary approach included for estimating the false negative rate. In my mind this would be simulations with known de novo mutations introduced into the offspring, run through the mapping, variant calling, filtering, and overlap steps to evaluate the best case scenario false negative rate.

Other concerns related to the false negative rate: Is there some circularity in defining quality thresholds in terms of sites that should be truly heterozygous and then also estimating the false negative rate from this? I am not entirely clear how this was applied from reading the text. Another reason I think simulations of the type described above are important is that sites that can be called with high confidence as homozygous for the alternative allele in one of the parents may be subtly non-random regions of the genome of (e.g. regions of lower diversity where a given read differs less from the reference sequence). As a related note, the authors should report what proportion of reads uniquely mapped to help evaluate this.

B) Number of background sites: The mutation rate estimate depends critically on the number of sites considered as the background set. I was unclear on how the authors determined this besides excluding repetitive regions and regions of low mappability (leaving them with ~440 Mb). The authors apply a set of filtering criteria, some of which apply only to variant sites (SNP quality) and some of which also should apply to invariant sites (depth, mapping quality). GATK also can provide an invariant quality score that can be used. My major question is whether the authors excluded any invariant sites from their background set when applying quality filters. Furthermore, the overlap approach should also be applied to invariant sites and in that sites that are called as variant in one of the two pipelines or fail an invariant quality metric in one the of pipelines should be excluded to make the SNP and background sites comparable. Otherwise variant sites will be disproportionately removed while the background number of sites held constant, potentially contributing to the low observed mutation rate. Both a simulation approach and including more detail in the manuscript will help to address this concern.

3) Investigation of how demographic history might contribute to low nucleotide diversity. Another possible cause of a low diversity besides the drift-barrier hypothesis is a recent bottleneck. Based on the group's previous work (Martinez Barrio et al. 2016) there is evidence for population size changes in herring, but I was unsure how to interpret the time scale and its likely effect on diversity given that the parameters used in this previous study were from humans. However, even if the authors redo these analyses with corrected parameters, I think this remains a concern since even modified PSMC methods like dical can miss demographic events in the very recent past that could affect diversity levels. In my reading the authors should either explore this possibility or significantly tone down their claims about the drift barrier hypothesis.

*Reviewer #3:*

The authors report a study in which the base-substitution mutation rate in the Atlantic herring is estimated from a two-family pedigree (two pairs of parents, six offspring per parental pair). The study is motivated by the low nucleotide diversity given large census size of the species, which leads to the suggestion that the mutation rate may be atypically low. Sure enough, the base-substitution rate reported is the lowest yet reported among vertebrates, although it's not a great deal lower. The methods seem sound, and I think their method for estimating the false negative rate is good. The results are credible, and the Discussion is sensible. It appears that indels were filtered out, but this should be noted explicitly in the Discussion.

[Editors' note: further revisions were requested prior to acceptance, as described below.]

Thank you for resubmitting your work entitled "Moderate nucleotide diversity in the Atlantic herring is associated with a low mutation rate" for further consideration at *eLife*. Your revised article has been favorably evaluated by Diethard Tautz (Senior Editor) and two reviewers, one of whom is a member of our Board of Reviewing Editors.

The manuscript has been improved but there are some remaining issues that need to be addressed before acceptance. Notably, both reviewers felt that there were still salient details missing from the Materials and methods and that the discussion of Lynch's hypothesis needed some revisions.

*Reviewer #1:*

I found the revised version much improved. My only major concern is that I believe that the per generation mutation rate is not correctly calculated. The authors do so by considering the number of mutations and dividing by the number of individuals, then the number of sites considered. But if mutation i is observed in K_i_ offspring (of those offspring that were fully sequenced), it should be counted K_i_ times. So the correct calculation, I believe, should be the sum over i of K_i_ divided by the number of offspring (and then number of sites).

I also think that the discussion of Lynch's hypothesis in the Discussion confuses census and effective population size. In Lynch's model, species with high *effective* (not census) population size have more effective selection on mutation modifiers and hence a lower equilibrium mutation rate. (Being a neutralist, he assumes that larger census population sizes have larger effective population sizes, but that need not be the case, for the reasons mentioned by the authors.) Thus, in the fourth paragraph of the Discussion, the authors should add "effective" to the two statements. More importantly, I do not see the basis for the claim that the low mutation rate helps to explain most of the discrepancy between diversity levels and census population size. According to the numbers provided by the authors, N is 10^[11]^ and N_e_ = 5x10^[5]^. Of these many orders of magnitude, the mutation rate helps with, at most, one.

Moreover, salient details are still missing for Methods to be fully reproducible. Notably, it is unclear how the authors introduce mutations in the reads and whether they do so mindful of other SNPs in reads (i.e., LD) or not, which might matter since they use a haplotype aware version of GATK. Also what does the cutoff for allelic balance (and other filters) end up being when a two SD criterion is imposed? This information is important for understanding which mosaic mutations are filtered out, among other reasons. I suggest reporting them in a supplementary table.

Finally, I wondered if something went wrong with the values reported in Table 3 checked one set of numbers by curiosity, the ones for collared flycatcher, and the reference cited has π = 3.3x10^-3^. When I divide that by (4x4.6x10^-9^), I get 1.8x10^-5^, the value provided for mouse, rather than the 4.9x10^[4]^ reported. Am I missing something? If not, it might be worth double checking the rest.

*Reviewer #2:*

I previously reviewed this paper and found the revision much improved. The questions remain relevant and interesting. I have a few additional substantive comments on the revision, outlined below:

1) Remaining missing information. The revised version of the paper does a much better job clearly outlining methods and making it easier for the reader to follow what was done. There are a few places with remaining issues. The false negative estimation simulations for example, are missing key details that would be needed to replicate the procedure. For example, what coverage was simulated, what ratio of reads supported each SNP, was quality score uniform or drawn from a distribution? Did the authors re-estimate quality distributions from the simulated data or apply the same filters they determined with the empirical data?

2) A paper recently posted on bioRxiv estimates the de-novo mutation rate in several cichlid fish species (http://biorxiv.org/content/early/2017/05/31/143859). This study is done in trios so confidence is lower, but results in a similar estimate of the mutation rate. This has important implications for several points in the authors' Discussion, particularly the interplay between mutation rate and diversity, as well as the temperature argument, and should be included in the Discussion. In addition, this is a stronger argument against the drift-barrier hypothesis and should be added to Discussion.

3) Statements about explanatory power. There are several summary statements in the Discussion that imply that the authors have directly evaluated the impact of various factors on π in herring. Most notably, the authors state: "In the case of the Atlantic herring, the low mutation rate, in conjunction with demographic history and efficient purifying selection, explains the majority of the apparent disparity between nucleotide diversity and the census population size in the Atlantic herring." The authors have not evaluated how different factors contribute so should not make statements such as this one.

---

## [Author Response]

*Given the enthusiasm for the central finding – of a lower mutation rate in a species with a large census population size – we would like to give you the opportunity to revise your manuscript and resubmit it for review. Please note, however, that reviewers are requesting both an additional analysis to confirm your estimated false negative rate and a thorough revision of the interpretation and discussion of the results. Specifically, in revising your manuscript, we ask that you address these following points:*

*1) Reviewers 1 and 2 raised important concerns about the lack of details provided in the Materials and methods. Moreover, they would like to see the false negative rate confirmed by a second approach (a number of possible options are suggested in their specific comments below).*

We have amended the Materials and methods section as suggested and we have confirmed the false negative rate using another approach (see below).

*2) Reviewers 1 and 2 felt that claims about the effective population size of herring are often incorrectly or confusingly framed. Notably, there is clear circularity in some sections, since N_e_ estimates from diversity rely on mutation rate estimates. This section needs to be thoroughly rewritten and, as reviewer 1 suggests, could be reframed in terms of the census population size instead. Indeed, Lewontin's paradox is about how similar diversity levels are across species with vastly different census population sizes.*

*This issue affects not only the interpretation of the findings in herring, but also the Discussion of findings of other organisms (Table 3). This Table reports N_e_ estimates and direct mutation rate estimates, but it is unclear if the N_e_ estimates were obtained using the direct estimates of mutation rates. The authors should really be starting with diversity levels, measured in a fairly uniform way across species, then using the direct mutation rate estimates to obtain long-term estimates of N_e_ – and only then examining the dependence of mutation rates with N_e_.*

The text concerning estimates of N_e_ has been completely revised and Table 3 much improved.

*3) Reviewers 1 and 2 also both doubted how much support the data really provide for Lynch's hypothesis, given the data for different species presented by the authors in Table 3 (even taken at face value).*

We have softened the text with the reference to Lynch’s hypothesis since we agree that we need more data from additional species before we can draw firm conclusions on this topic.

*Reviewer #1:*

*I find the central result – the estimated mutation rate per generation-to be of great interest, even though I have a concern about the estimated false negative that I would like to see addressed (see 1 below). I also found the results about mosaicism striking and think they warrant further emphasis in the paper. However, I found the discussion of effective population sizes very confusing, and suggest that this section be removed or thoroughly rewritten. I am also unconvinced by how clearly the data support Lynch's hypothesis, when considered together with other data (presented in Table 3).*

*1) Given that the mutation rate reported by the authors is lower than ever report for another multi-cellular organism, a reliable estimate of the false negative is critical. I found the approach taken by the authors sensible, but would like to see a second approach to be convinced by the finding that it is indeed very low. Options include relaxing the threshold and verifying these additional mutation calls, to ensure that few validate, or flipping alleles in the reads to simulate new mutations, as done for example by Venn et al. 2014 Science.*

We have followed this advice and estimated the false negative rate by an approach similar to Venn et al. (see Materials and methods) This gives an estimated false negative rate that is similar to the one obtained previously. The results are presented in [Supplementary-material SD1-data] and summarized in the text.

*In this regard, I note that the authors were only able to assess mutations in roughly half of the genome. That is not a problem in itself, but it does mean that they are making a strong assumption that the mutation rate in the other half of the genome is the same. In this regard, it would be helpful to know whether diversity levels are lower in the fraction of the genome surveyed. Regardless of the answer, a caveat is needed about the rest of the genome and what it may yet reveal.*

We discuss this concern in much more detail now in the second paragraph of the Discussion. We notice that the estimated nucleotide diversity in fact is higher in the part of the genome that was used in this study compared with the part that was excluded. This is most likely explained by the fact that SNP calling is hampered in regions with low mappability due to the presence of repeats. We note that the mutation rate may be higher in repeat regions but also argue this concern also apply to the data from other species because it is very challenging to accurately make SNP calling in repeat regions.

*On a more minor note, I would like to know how many sites were used to estimate the false negative rate by their approach.*

It was 116,910 sites. This is now given in the text.

*2) Quite confusingly, the authors refer repeatedly to an estimate of N_e_ obtained "from coalescent analyses" without mentioning that this estimate depends entirely on assumptions about the mutation rate. From what I gathered looking at the previous paper, the authors used the human per generation rate for that analysis (?!), suggesting that they should actually be reporting 7x that number for N_e_. Moreover, as best as I can understand, they are referring to the estimate of N_e_ at present, not the long-term estimate. Yet neutral diversity levels (i.e., the 0.3% per site that they cite) represent much longer time periods (4N_e_ generations on average in a constant population size), and they estimate huge fluctuations in N_e_ over that time scale in their previous paper.*

We agree that this was confusing and we have completely revised this text.

*In fact, the only information that they have about N_e_ that does not depend on the mutation rate is that LD decays quickly (similar to the scale of Drosophila, not humans), which could be due to a high rate of recombination, and that Fst is low, which could be due to high migration. Thus, I would argue that they don't really know if N_e_ "should be high", as argued throughout.*

*What they do know, and in my mind do not discuss enough, is that the census population size is surely huge, on the order of billions per school. So the mystery is how N_e_ can be only 0.3x10^-2^/(4x1.7x10^-9^) =4.4x10^[5]^ when the census population size is probably on the order of 10^[10]^ or more. That is in my view quite a striking result, but needs to be recast as such.*

We agree and the Discussion of this has been improved thanks to these valuable comments.

*3) Similarly, in Table 3, the authors present a set of N_e_ estimates without stating whether these are obtained assuming the direct estimates of mutation rates or not. With regard to Lynch's hypothesis, these data are not overwhelmingly supportive, other than in suggesting that mammals, who are fewer in number at least, have a slightly higher mutation rate. In part, I would argue that reflects the limitations of N_e_ as a proxy for the efficiency of selection, since it has different meanings in inbreeding species than outcrossers, and structured populations than panmictic ones. More striking in my mind is how similar rates are across very diverse organisms.*

This has now been amended and clarified in the revised version.

*As an aside, the existing mice estimates are missing (http://biorxiv.org/content/early/2016/10/20/082297 and Ishimura et al. 2015 Genome Research.*

These references have been added.

*Additional comments:*

*The authors might add a p-value to the statement that the male bias is less than in mammals (e.g., assuming α = 3, I obtain p=0.01).*

We have now added such a statement to the text.

*In discussing the fraction of CpG transitions, it may be helpful to add a comment about whether CpGs are methylated in teleosts.*

We have now pointed out that CpGs are methylated in teleosts and added a citation on this topic.

*Are there estimates of the mutation rate based on the number of substitutions between species? How do they compare to the estimate presented here?*

This discussion is out of the scope of the present study and is better suited for a review on the relationship between substitution rates and mutation rates.

*Table 2 suggests apparent difference in the degree of mosaicism between the two families (e.g., I obtain p = 0.04 by a FET). The authors should comment on this point.*

We now highlight this difference but do not discuss it further since we only have data on two families so the biological significance is unclear.

*Reviewer #2:*

*Feng et al. provide the first direct estimation of mutation rate in fish and bring up the possibility that the low mutation rate they observe is directly related to the low nucleotide diversity in herring, despite its incredibly large population size. I find this study of great interest in expanding our understanding of mutation rates and their variation but had some issues in how the authors interpreted their data, methodological approaches and the level of detail provided in the manuscript. These are outlined below:*

*1) The manuscript contains too little detail on approach to evaluate the methods. Several filters are referenced in the text such as sequence depth, SNP and genotype quality statistics, but no information is given on what these parameters were. The authors state that the cutoffs were based on SNPs that were fixed for different alleles in both parents but it is unclear how – did the authors use the mean values from this distribution, the minimum? Furthermore, the authors state that they follow the GATK pipeline to generate variant calls but there are different paths through this pipeline depending on the specifics of the data, for example, whether known variants are available. Information about library preparation (e.g. confirming that libraries were PCR-free) would also be useful. In my view, the manuscript cannot be appropriately evaluated without much of this missing information. As a minor note, I looked up the SRA accession to try to get more detail on the library preparation and could not find it on the SRA (perhaps the link is still private?).*

The Materials and methods section has been updated so that our pipeline is described in more detail. We have added the information that the sequencing libraries were PCR-free and the SRA accession will be opened when the paper is published.

*2) Evaluation of the false negative and number of background sites. The authors use a sensible approach to evaluate their false negative rate among SNPs by focusing on sites that are homozygous for alternative alleles in both parents and asking how frequently these are detected as heterozygous in the offspring. However, I have several questions/concerns about the false negative rate and number of background sites.*

*A) False negative rate: Because the false negative rate is a fundamentally important in the major result of this study, I would like to see a secondary approach included for estimating the false negative rate. In my mind this would be simulations with known de novo mutations introduced into the offspring, run through the mapping, variant calling, filtering, and overlap steps to evaluate the best case scenario false negative rate.*

This has now been done resulting in similar results (see text).

*Other concerns related to the false negative rate: Is there some circularity in defining quality thresholds in terms of sites that should be truly heterozygous and then also estimating the false negative rate from this? I am not entirely clear how this was applied from reading the text. Another reason I think simulations of the type described above are important is that sites that can be called with high confidence as homozygous for the alternative allele in one of the parents may be subtly non-random regions of the genome of (e.g. regions of lower diversity where a given read differs less from the reference sequence). As a related note, the authors should report what proportion of reads uniquely mapped to help evaluate this.*

We do not think our approach is circular and as indicated above a simulation study gave a similar result that in turn results in a very similar estimated mutation rate when corrected for the false negative rate.

*B) Number of background sites: The mutation rate estimate depends critically on the number of sites considered as the background set. I was unclear on how the authors determined this besides excluding repetitive regions and regions of low mappability (leaving them with ~440 Mb). The authors apply a set of filtering criteria, some of which apply only to variant sites (SNP quality) and some of which also should apply to invariant sites (depth, mapping quality). GATK also can provide an invariant quality score that can be used. My major question is whether the authors excluded any invariant sites from their background set when applying quality filters. Furthermore, the overlap approach should also be applied to invariant sites and in that sites that are called as variant in one of the two pipelines or fail an invariant quality metric in one the of pipelines should be excluded to make the SNP and background sites comparable. Otherwise variant sites will be disproportionately removed while the background number of sites held constant, potentially contributing to the low observed mutation rate. Both a simulation approach and including more detail in the manuscript will help to address this concern.*

We believe that the depth criterion is the most relevant here, as all high-mappability regions should have similar mapping quality distributions. We have now estimated that 2.6% of the invariant sites would fail the depth cutoff. The text now includes this, and the final rates and CIs have been adjusted accordingly.

*3) Investigation of how demographic history might contribute to low nucleotide diversity. Another possible cause of a low diversity besides the drift-barrier hypothesis is a recent bottleneck. Based on the group's previous work (Martinez Barrio et al. 2016) there is evidence for population size changes in herring, but I was unsure how to interpret the time scale and its likely effect on diversity given that the parameters used in this previous study were from humans. However, even if the authors redo these analyses with corrected parameters, I think this remains a concern since even modified PSMC methods like dical can miss demographic events in the very recent past that could affect diversity levels. In my reading the authors should either explore this possibility or significantly tone down their claims about the drift barrier hypothesis.*

We agree and we have toned down the drift barrier hypothesis.

*Reviewer #3:*

*The authors report a study in which the base-substitution mutation rate in the Atlantic herring is estimated from a two-family pedigree (two pairs of parents, six offspring per parental pair). The study is motivated by the low nucleotide diversity given large census size of the species, which leads to the suggestion that the mutation rate may be atypically low. Sure enough, the base-substitution rate reported is the lowest yet reported among vertebrates, although it's not a great deal lower. The methods seem sound, and I think their method for estimating the false negative rate is good. The results are credible, and the Discussion is sensible. It appears that indels were filtered out, but this should be noted explicitly in the Discussion.*

We did not include indels because previous studies have indicated that the mutation rate for InDels is about an order of magnitude lower than for point mutations (see e.g. Besenbacher et al. Multi-nucleotide de novo mutations in humans. 2016. PLoS Genetics). It is likely that this is the case also in herring because the rate of InDels is about an order of magnitude lower than SNPs. Thus, our material is too small to generate meaningful estimates of the mutation rate for InDels in the herring. In the Introduction we now state: “we estimate the genome-wide point mutation rate in Atlantic herring.”

[Editors' note: further revisions were requested prior to acceptance, as described below.]

*The manuscript has been improved but there are some remaining issues that need to be addressed before acceptance. Notably, both reviewers felt that there were still salient details missing from the Materials and methods and that the discussion of Lynch's hypothesis needed some revisions.*

*Reviewer #1:*

*I found the revised version much improved. My only major concern is that I believe that the per generation mutation rate is not correctly calculated. The authors do so by considering the number of mutations and dividing by the number of individuals, then the number of sites considered. But if mutation i is observed in K_i_ offspring (of those offspring that were fully sequenced), it should be counted K_i_ times. So the correct calculation, I believe, should be the sum over i of K_i_ divided by the number of offspring (and then number of sites).*

We agree and have recalculated the mutation rate accordingly. The estimated mutation rate changed from 1.7 x 10^-9^ to 2.0 x 10^-9^.

*I also think that the discussion of Lynch's hypothesis in the Discussion confuses census and effective population size. In Lynch's model, species with high effective (not census) population size have more effective selection on mutation modifiers and hence a lower equilibrium mutation rate. (Being a neutralist, he assumes that larger census population sizes have larger effective population sizes, but that need not be the case, for the reasons mentioned by the authors.) Thus, in the fourth paragraph of the Discussion, the authors should add "effective" to the two statements. More importantly, I do not see the basis for the claim that the low mutation rate helps to explain most of the discrepancy between diversity levels and census population size. According to the numbers provided by the authors, N is 10^[11]^ and N_e_ = 5x10^[5]^. Of these many orders of magnitude, the mutation rate helps with at most one.*

We have changed population size to effective population size in the third paragraph of the Discussion. We did not write that the low mutation rate explains most of the discrepancy, we wrote that the mutation rate together with demographic history and efficient selection explain most of the discrepancy, but we have rephrased this sentence.

*Moreover, salient details are still missing for Methods to be fully reproducible. Notably, it is unclear how the authors introduce mutations in the reads and whether they do so mindful of other SNPs in reads (i.e., LD) or not, which might matter since they use a haplotype aware version of GATK. Also what does the cutoff for allelic balance (and other filters) end up being when a two SD criterion is imposed? This information is important for understanding which mosaic mutations are filtered out, among other reasons. I suggest reporting them in a supplementary table.*

The mutated reads were not selected taking surrounding SNPs into account, but rather a random subset of the reads overlapping the chosen site were mutated for the relevant base, with the frequency of mutated reads for each synthetic mutation being sampled from the distribution of empirically identified heterozygous sites. Due to the haplotype-aware nature of HaplotypeCaller, ignoring the context of local variation, as we did, could, if anything, induce a slight over-estimation of the false negative rate.

We were aiming to detect transmitted parental mutations, not ones that are mosaic in the offspring, which is why we based the cutoff on known, inherited heterozygous sites. For all quality values, we filtered with the cut-offs derived from the original data set. The main cut-offs were minor allele frequency >= 0.25, depth >= 20 and GQ = 99.

*Finally, I wondered if something went wrong with the values reported in Table 3 checked one set of numbers by curiosity, the ones for collared flycatcher, and the reference cited has π = 3.3x10^-3^. When I divide that by (4x4.6x10^-9^), I get 1.8x10^-5^, the value provided for mouse, rather than the 4.9x10^[4]^ reported. Am I missing something? If not, it might be worth double checking the rest.*

We have corrected these mistakes. Thanks for spotting this.

*Reviewer #2:*

*I previously reviewed this paper and found the revision much improved. The questions remain relevant and interesting. I have a few additional substantive comments on the revision, outlined below:*

*1) Remaining missing information. The revised version of the paper does a much better job clearly outlining methods and making it easier for the reader to follow what was done. There are a few places with remaining issues. The false negative estimation simulations for example, are missing key details that would be needed to replicate the procedure. For example, what coverage was simulated, what ratio of reads supported each SNP, was quality score uniform or drawn from a distribution? Did the authors re-estimate quality distributions from the simulated data or apply the same filters they determined with the empirical data?*

We used the empirical coverage for each selected site, and the allele balance for each site was sampled from the allele distribution of known heterozygous sites. The decision about which reads to mutate was uninformed with regard to any quality measure, and after inserting a mutation we kept the quality values from the original reads. We applied the same filters as for the original data set and did not re-estimate the cut-offs based on the simulated reads, as they were from a non-random (i.e. high-mappability) subset of the genome.

*2) A paper recently posted on bioRxiv estimates the de-novo mutation rate in several cichlid fish species (http://biorxiv.org/content/early/2017/05/31/143859). This study is done in trios so confidence is lower, but results in a similar estimate of the mutation rate. This has important implications for several points in the authors' Discussion, particularly the interplay between mutation rate and diversity, as well as the temperature argument, and should be included in the Discussion. In addition, this is a stronger argument against the drift-barrier hypothesis and should be added to Discussion.*

We now cite this paper in the fourth paragraph of the Discussion. The fact is that the estimated mutation rate is higher than in Atlantic herring (3.5x10^-9^ vs. 2.0x10^-9^). But the data are still not conclusive since the estimated effective population size is significantly smaller than herring but they also live in warmer waters. We do not want to discuss the cichlid data in too much detail at this stage since it is based on only three trios and the paper has not yet been peer reviewed.

*3) Statements about explanatory power. There are several summary statements in the Discussion that imply that the authors have directly evaluated the impact of various factors on π in herring. Most notably, the authors state: "In the case of the Atlantic herring, the low mutation rate, in conjunction with demographic history and efficient purifying selection, explains the majority of the apparent disparity between nucleotide diversity and the census population size in the Atlantic herring." The authors have not evaluated how different factors contribute so should not make statements such as this one.*

We have rephrased this sentence and conclude that both low mutation rate and demographic history contribute to explaining the rather low nucleotide diversity, without inferring the relative impact of different factors.